# Influence of Thermocompression Conditions on the Properties and Chemical Composition of Bio-Based Materials Derived from Lignocellulosic Biomass

**DOI:** 10.3390/ma17081713

**Published:** 2024-04-09

**Authors:** Julie Cavailles, Guadalupe Vaca-Medina, Jenny Wu-Tiu-Yen, Jérôme Peydecastaing, Pierre-Yves Pontalier

**Affiliations:** 1Laboratoire de Chimie Agro-Industrielle (LCA), Université de Toulouse, INRAE, Toulouse INP, 31030 Toulouse, France; julie.cavailles@hotmail.com (J.C.); guadalupe.vacamedina@toulouse-inp.fr (G.V.-M.); jerome.peydecastaing@toulouse-inp.fr (J.P.); 2eRcane, Sainte-Clotilde, 97490 La Réunion, France; jenny.wutiuyen@ercane.re

**Keywords:** thermocompression, binderless materials, lignocellulosic biomass, bio-based materials, mechanical properties, water resistance

## Abstract

The aim of this study was to assess the influence of thermocompression conditions on lignocellulosic biomasses such as sugarcane bagasse (SCB) in the production of 100% binderless bio-based materials. Five parameters were investigated: pressure applied (7–102 MPa), molding temperature (60–240 °C), molding time (5–30 min), fiber/fine-particle ratio (0/100–100/0) and moisture content (0–20%). These parameters affected the properties and chemical composition of the materials. The density ranged from 1198 to 1507 kg/m^3^, the flexural modulus from 0.9 to 6.9 GPa and the flexural strength at breaking point from 6.1 to 43.6 MPa. Water absorption (WA) and thickness swelling (TS) values ranged from 21% to 240% and from 9% to 208%, respectively. Higher mechanical properties were obtained using SCB with fine particles, low moisture content (4–10%) and high temperature (≥200 °C) and pressure (≥68 MPa), while water resistance was improved using more severe thermocompression conditions with the highest temperature (240 °C) and time (30 min) or a higher moisture content (≥12.5%). Correlations were noted between the mechanical properties and density, and the material obtained with only fine particles had the highest mechanical properties and density. Material obtained with a 30 min molding time had the lowest WA and TS due to internal chemical reorganization followed by hemicellulose hydrolysis into water-soluble extractables.

## 1. Introduction

The use of lignocellulosic by-products from the processing of agricultural resources is gaining ground in the industry as they are inexpensive, highly available, renewable and biodegradable. Lignocellulosic resources are promising for the production of eco-friendly and value-added materials. They are used for reinforcement in polymer-based composites [1,2], or to produce disposable tableware [3], food packaging [4] and particleboard [5,6,7]. However, pulp production is involved in the manufacturing of these materials, which also requires considerable chemical treatments [8] or the addition of synthetic resins [9]. The use of these chemicals may have a negative impact by giving rise to environmental and health issues. Currently, urea-formaldehyde resin is widely used as binder in the production of particleboard and its use can lead to formaldehyde and volatile organic compound (VOC) emissions from the finished materials, representing a hazard to human health [10,11]. Research on the production of binderless materials, to avoid the use of any chemical additives, has hence gained interest.

Uniaxial thermocompression is a recognized process for the fabrication of materials that uses high pressure and temperature to mold the materials into the desired shape. Cohesive lignocellulosic materials can be produced without using binders or any chemical agents. Several self-bonding mechanisms of binderless lignocellulosic materials have been proposed as consolidation mechanisms [12], such as lignin-carbohydrate complex (LCC) crosslinking, hydrogen bonding and cellulose co-crystallization. Self-bonding of this kind of material is mainly attributed to the presence of lignin, which is considered to be partially responsible for material adhesion [13]. Lignin has been shown to serve as a natural binder through its plasticization, which occurs under the high pressure and temperature applied during thermocompression, resulting in cohesive and self-bonded materials. In addition, mechanisms involving cellulose, hemicellulose and lignin degradation under thermocompression have been described [14], inducing self-polymerization and crosslinking upon further condensation reactions between furfural and lignin, in turn contributing to self-bonding [15,16]. Cohesive binderless boards have been successfully produced through the thermocompression of lignocellulosic fibers from sunflower cake [17], wheat straw [18], *Miscanthus sinensis* [19], banana bunches [20], kenaf core [21], coriander straw [22], rice straw [23], amaranth stems [24], oil palm trunks [25] and sugarcane bagasse [26]. However, binderless lignocellulosic materials exhibit greater sensitivity to water than materials obtained with binders. It was demonstrated that thickness swelling of flax shive particleboard was at least doubled in the absence of binder [27].

In order to manufacture binderless materials with improved mechanical properties and water resistance compared to materials with binders, thermocompression conditions are important and need to be adapted as they directly influence the material properties. Many researchers have studied the effects of temperature [28], time [29] and pressure [30] on the density, mechanical properties and water resistance of lignocellulosic binderless materials [17,31]. Other factors, such as particle size [32] and moisture content [21], can also have an impact on the material properties. However, few studies have focused on linking the effects of thermocompression conditions on the chemical composition of lignocellulosic materials and their properties [15,16]. Sugarcane bagasse is one of the most interesting lignocellulosic biomasses for this purpose, given that sugarcane is a major agricultural crop, with a global production of around 1967 billion tons in 2021 [33].

The aim of this study was to determine the effects of thermocompression conditions on the properties and chemical composition of binderless sugarcane bagasse-based materials.

## 2. Materials and Methods

### 2.1. Raw Material Preparation 

Air-dried sugarcane bagasse (SCB) was provided by eRcane (Réunion, France). SCB was ground into a 2-mm grid using an MF 10 basic Microfine grinder drive from IKA-Werke (Staufen im Breisgau, Germany). A portion of ground SCB was divided using an AS 200 Basic vibratory sieve shaker from Retsch (Haan, Germany) for 10 min at 1 mm amplitude using a series of three sieves with mesh openings of 1.0 mm, 0.8 mm, 0.2 mm and a bottom plate. Two fractions—with fibers retained between the 0.2 and 0.8 mm sieves, and fine particles retained below 0.2 mm—were obtained, at a ratio of 65/35, respectively, from ground SCB. The two fractions were mixed with different proportions to modify the fiber/fine-particles ratio and study its effect. Ground SCB and these two fractions were conditioned at 25 °C and 50% relative humidity (RH) for 3 weeks prior to thermocompression.

### 2.2. Analytical Methods

Chemical compositions of ground SCB and these two initial fractions, as well as all SCB materials, were determined in triplicate. The contents are expressed as a percentage of dry matter (DM). The DM content was assessed after drying at 105 °C until constant weight, and the ash content was determined after mineralization at 550 °C for 12 h [34]. Chemical characterization was done using a procedure based on the laboratory analytical procedure of the National Renewable Energy Laboratory (NREL) [35]. The NREL protocol was adapted for fine particles, with 40 min of acidic hydrolysis instead of 60 min to reduce the amount of degradation products obtained during hydrolysis. This procedure, which is suitable for extractable-free biomass, required prior determination of water and ethanol-soluble molecules following water and then ethanol (96%) extractions. Extractions were conducted using 1 g of initial sample and 100 mL of boiled solvent for 1 h on a Fibertech FT 122 extraction system from Foss (Hillerød, Denmark).

Cellulose, hemicellulose and lignin contents were determined after a two-step hydrolysis process using 72% sulfuric acid from VWR (Radnor, PA, USA) at 30 °C for 1 h, and then a 4% sulfuric acid solution after dilution with deionized water at 121 °C for 1 h, followed by filtration. Acid soluble lignin (ASL) content was determined in the liquid fraction on a UV-1800 spectrophotometer from Shimadzu (Kyoto, Japan) at 240 nm using an absorptivity constant of 25 L/g·cm, while the acid insoluble lignin (AIL) content was measured gravimetrically after calcination of the solid residue at 450 °C for 12 h. The liquid fraction was neutralized with calcium carbonate from Merck (Darmstadt, Germany) until reaching neutral pH, and then filtered on a 0.2 µm cellulose acetate filter before sugar monomer (i.e., arabinose, glucose and xylose) and sugar degradation product (i.e., acetic acid, furfural and HMF) analysis performed on a Thermo Ultimate 3000 HPLC system from Dionex (Sunnyvale, CA, USA). All standards were purchased from Sigma-Aldrich (Saint-Louis, MO, USA). A Rezex RHM-Monosaccharide H+ 300 × 7.8 mm column connected to a Rezex RHM-Monosaccharide H+ 50 × 7.8 mm guard column, both from Phenomenex (Torrance, CA, USA), was used with 5 mmol/L H_2_SO_4_ as eluent at 0.6 mL/min. The injection volume was 50 µL, while the column was maintained at 65 °C and the RI detector at 50 °C.

### 2.3. Physical and Thermal Characterization of SCB

The bulk and tapped densities of ground SCB and its two initial granulometric fractions were determined using a Densitap ETD-20 volumenometer from Granuloshop (Chatou, France) fitted with a 250 mL graduated cylinder. The sample was weighed in the graduated cylinder and the volume was recorded just before compaction to determine the bulk density. The cylinder was then tapped 500 times on the volumenometer, at 3 mm height and 250 taps/min speed. The volume was measured at the nearest graduation and the operation was then repeated until a constant volume was obtained to determine the tapped density. All measurements were performed in triplicate.

The particle-size distributions and the estimated specific surface area (SSA) of the ground SCB and its two granulometric fractions were determined using a Mastersizer 3000 laser-diffraction particle-size analyzer from Malvern Panalytical (Palaiseau, France) in solid mode. The particle-size distributions generated were characterized using D10, D50 and D90, indicating the size below which 10%, 50% and 90% of the particles are found under the series of sieves based on the cumulative distribution.

Optical and SEM images of the two granulometric fractions from ground SCB (fiber and fine particles) were respectively obtained using a Nikon SMZ1500 binocular loupe (Tokyo, Japan) and a FEI Quanta 450 scanning electron microscope (Hillsboro, OR, USA), with 130 Pa water-vapor partial pressure in the chamber at high voltage (12.5 kV) without saturation of the samples.

Thermogravimetric analysis (TGA) was performed to determine the thermal-degradation temperature of SCB and the derivative thermogravimetry (DTG). About 10 mg of ground SCB was placed in a 70 µL alumina crucible and analyzed on a thermogravimetric analyzer from Mettler-Toledo (Columbus, OH, USA). The analysis was conducted under a 20 mL/min airflow with a 25 °C to 550 °C temperature scan at a 5 °C/min heating rate.

### 2.4. Uniaxial Thermocompression

All SCB materials were obtained by thermocompression of 20 g of SCB using a steel mold. A 50 t capacity heated hydraulic press from Pinette Emidecau Industries (Chalon-sur-Saône, France) was used to produce flat square materials measuring 70 mm × 70 mm. Five variables were investigated: molding pressure (7, 17, 34, 68 and 102 MPa), molding temperature (60, 80, 100, 120, 140, 160, 180, 200, 220 and 240 °C), molding time (5, 10, 15, 20 and 30 min), fiber/fine-particle ratio (0/100, 25/75, 50/50, 65/35, 75/25 and 100/0) and MC (0, 4.2, 7.2, 10.4, 12.4, 15.4, 16.8 and 20.0%). SCB materials were manufactured under 30 different conditions, as presented in Table 1. For the fixed parameters, molding pressure was set at the maximum pressure achievable with the press used for a material measuring 70 mm × 70 mm at a value of 102 Mpa. Molding temperature and time were set at 180 °C and 5 min, respectively, to limit the thermal degradation of the material while ensuring sufficient cohesion. The fiber/fine-particle ratio was set at 65/35, which corresponded to the initial ratio of the ground SCB used, while the SCB moisture content was set at 7.2%, since this value corresponded to the moisture content of the ground SCB obtained after equilibration at 25 °C and 50% relative humidity for 3 weeks prior to thermocompression.

The thermocompression cycles ranged from 17 to 45 min, depending on the molding time, according to the following procedure: uniaxial pressure was applied to the mold, then the mold was heated to the desired temperature. The temperature was maintained for the desired time and, finally, the mold was cooled down while maintaining the pressure before opening. For each tested molding condition, two materials were produced and pictures were taken. The materials were each then cut into eight 45 mm long and 10 mm wide specimens; finally, these were stored in an environmental chamber at 25 °C and 50% RH for 2 weeks until constant weight before testing the properties. A small piece of material from each molding condition was ground into a 2 mm grid before applying the NREL protocol to characterize its chemical composition. 

### 2.5. Density Measurement of the Materials

The density of the materials after thermocompression was determined using the remaining pieces of material after cutting the bending test specimens in triplicate. The densities of these samples were assessed using a method based on Archimedes’ principle, with cyclohexane as the immersion liquid. When a solid is immersed in a liquid, it experiences a force known as buoyancy. The value of this force is equal to the weight of the volume of liquid displaced by the sample. Using a Sartorius hydrostatic balance (Göttingen, Germany) capable of weighing in both air and liquid, we were able to determine the density of our samples with the following Formula (1):(1)$$\mathrm{Density}=\frac{w_{\mathrm{air}}*\left(d_{\mathrm{cyclohexane}}-d_{\mathrm{air}}\right)}{\left(w_{\mathrm{air}}-w_{\mathrm{cyclohexane}}\right)*\mathrm{Corr}}+d_{\mathrm{air}}$$
where $w_{\mathrm{air}}$ and $w_{\mathrm{cyclohexane}}$ are the sample weights measured in air and cyclohexane, $d_{\mathrm{air}}$ and $d_{\mathrm{cyclohexane}}$ are the densities of air and cyclohexane at room temperature and Corr is the thrust correction factor due to the submerged wire.

### 2.6. Mechanical Bending Properties

The bending properties of the test samples were assessed using a Tinius Olsen universal testing machine (Horsham, PA, USA), equipped with a 500 N load cell, and the three-point bending test. The sample thicknesses and widths were measured at the center with a Tacklife electronic digital sliding caliper (Levittown, NY, USA). The testing speed was set at 1 mm/min, with a 40 mm grip separation. The loading direction was perpendicular to the upper surface of the sample, and the load was applied equidistant to the supports. All working conditions were tested by measuring the sixteen specimens cut for each molding condition. The evaluated properties were the flexural modulus and flexural strength at breaking point:(2)$$\mathrm{Flexural}\;\mathrm{modulus}=\frac{L^{3}}{4{\;*\;w*\;t}^{3}}*\;\frac{F_{2}-F_{1}}{d_{2}-d_{1}}$$
(3)$$\mathrm{Flexural}\;\mathrm{strength}=F_{\mathrm{breaking}}*\frac{3\;*\;L}{2{\;*\;w*\;t}^{2}}$$
where L is the length between the two supports, w is the sample width, t is the sample thickness, F_2_ and F_1_ are the forces measured for the deformations d_2_ and d_1_ and F_breaking_ is the force measured at breaking point.

### 2.7. Water Resistance Properties

Water resistance was tested by immersing 45 mm long and 10 mm wide specimens in water at 25 °C to determine the water absorption (WA) and thickness swelling (TS) in triplicate. Before soaking, the samples were oven-dried at 105 °C until constant weight to ensure a uniform initial condition. The initial thickness and weight were then measured. The samples were subsequently submerged in distilled water at 25 °C for 24 h. Weight and thickness measurements were taken hourly for the first 8 h and again after 24 h. The thickness of each sample was measured at three points, i.e., at the center and both ends. WA and TS were then calculated for each sample at 24 h using the following formulas:(4)$$\mathrm{WA}=\frac{w_{24h}-w_{0}}{w_{0}}*100$$
(5)$$\mathrm{TS}=\frac{t_{24h}-t_{0}}{t_{0}}*100$$
where w_0_ and w_24h_ are the sample weight initially and after 24 h of water immersion, t_0_ and t_24h_ are the sample thickness initially and after 24 h of water immersion.

### 2.8. Statistical Analysis

WA, TS and density determinations were conducted in triplicate. For the mechanical properties (flexural modulus and strength at breaking point), sixteen samples were tested for each condition. Means were statistically compared with a one-way analysis of variance (ANOVA) with α = 0.05 and a Student’s *t* test. Values with no significant difference are presented with the same letter (a–d). Regression equations were determined when R^2^ was higher than 0.85.

## 3. Results and Discussion

### 3.1. Characterization of Raw Sugarcane Bagasse

The physical characteristics of the ground SCB and its two granulometric fractions are presented in Table 2. The fine-particle fraction had higher bulk and tapped densities compared to the fiber fraction. This fraction compacted more easily than the fiber fraction, resulting in lower porosity. Ground SCB had bulk and tapped densities similar to those of the fine-particle fraction, showing that the fine particles in the ground SCB filled the pore spaces left by the fibers. The fine-particle fraction had a higher SSA of 98 m^2^/g as compared to 15 m^2^/g for the fiber fraction, while the SSA of ground SCB was proportional to its fiber/fine-particle ratio (65/35). 

The particle-size distributions of ground SCB, the fiber fraction and the fine-particle fraction are given in Figure 1a. The fine-particle fraction had particle sizes ranging from 1 to 600 µm, with a majority of particles between 34 and 161 µm, while the fiber fraction had particle sizes ranging from 25 to 3100 µm, with a majority of particles between 226 and 1210 µm. Ground SCB was heterogeneous but had a particle-size distribution, with D50 and D90, closer to that of the fiber fraction, given that fibers represented 65% of the ground SCB. A shoulder on the curve of the particle-size distribution of ground SCB was noted between 10 and 200 µm, corresponding to the fine-particle contribution. Determination of the aspect ratio (L/D), as shown in Figure 1b, based on optical images obtained with a binocular lens (Figure 2a,b), showed that the particles in the fine-particle fraction predominantly had an aspect ratio of between 1 and 2 (70%), with rather rounded shapes, and a few fibers with an aspect ratio of up to 6. The fiber fraction was predominantly made up of fibers, with an aspect ratio of up to 20, and it also included a small fraction of round particles, with an aspect ratio of between 1 and 2 (15%).

SEM images of the fiber and fine-particle fractions are presented in Figure 2. In agreement with the particle-size distribution, the fiber fraction had larger particles and fibers than the fine-particle fraction. The fiber fraction was predominantly made up of fibers from at least 1 mm to several millimeters long, but also contained spherical particles with a honeycomb structure (Figure 2e), characteristic of bagasse pith [36]. The fine-particle fraction contained more spherical particles of various shapes, along with short fibers <500 µm (Figure 2f).

The chemical compositions of the ground SCB and its two granulometric fractions used are presented in Table 3. Ground SCB consisted mainly of cellulose (36%), lignin (25%) and hemicellulose (20%). The hemicellulose fractions were mainly composed of xylose (90%) and a small proportion of arabinose (10%). Hemicellulose chains could therefore be made up of xylose units with some branching containing arabinose units. The chemical composition obtained for ground SCB was consistent with that reported in previous studies on SCB from Réunion [37] and the mass balance was close to 100% for all SCB fractions. Fine particles and fibers exhibited different cellulose, hemicellulose, water-soluble extractable and ash contents. Fine particles had lower cellulose and hemicellulose contents and higher ash and water-extractable contents compared to fibers. Fine particles were likely derived from parenchyma cells that may have still contained mineral components due to possible incomplete lysis during sugarcane milling prior to the juice extraction step. Larger particles were more likely structural fibers, as shown in the SEM images (Figure 2e,f), consisting mainly of lignocellulosic components. Ground SCB, consisting of 65% fibers and 35% fine particles, had a chemical composition representative of the mixture of these two fractions.

### 3.2. Properties and Chemical Composition of Binderless SCB Materials

#### 3.2.1. Effect of Molding Pressure

All materials made with different molding pressures had similar appearances (Figure 3). Their chemical compositions, as presented in Figure 4, were comparable with each other and with that of ground SCB. The pressure modification, therefore, did not lead to any change in chemical composition.

The variations in material density, mechanical properties and water resistance with the molding pressure are shown in Figure 5. The material density increased with molding pressure until a maximum density of around 1460 kg/m^3^ was reached at ≥68 MPa. These densities are higher than those of other SCB materials described in the literature [38] due to the application of a significantly higher pressure in our study (7–102 MPa) than what was used by others during SCB thermocompression [39]. Higher pressure could promote better contact between particles [29] and the materials obtained could occupy less volume with considerably higher densities within a narrow 1.4 to 1.5 g/m^3^ range [12]. The pressure and density followed a logarithmic curve with a density limit of around 1400 kg/m^3^ [40], in accordance with our results. The flexural modulus increased from 0.9 to 5.3 GPa, while the flexural strength increased with molding pressure from 6.1 to 37.6 MPa. Nevertheless, a plateau was reached, above 68 MPa pressure, in line with the trend noted with regard to the material density. The density and mechanical properties were clearly dependent and, due to the limit density value, the molding pressure modifications at high pressures had a limited effect on the material properties [41]. With regard to water resistance, materials prepared at 7 MPa molding pressure were broken prior to the end of the 24 h test in water. It seems that the pressure was too low to ensure cohesion of the material in water. For the other materials, WA ranged from 116% to 151% whereas TS ranged from 121% to 144%, which was relatively high compared to the properties of lignocellulosic materials formed with binder [42]. The poor properties of binderless lignocellulosic materials in contact with water is mainly due to the presence of polar groups, which attract water molecules through hydrogen bonding [9]. This phenomenon led to moisture build-up in the fiber cell wall, which was responsible for the material dimensional changes. However, WA and TS tended to decrease with increasing molding pressure up to 68 MPa. This could have been related to the increase in material density, which was previously described [43,44] and found to result in a reduction in the internal void volume [45].

#### 3.2.2. Effect of Molding Temperature

By varying the molding temperature between 60 and 240 °C, it was possible to reach the SCB thermal-degradation onset temperature during thermocompression. According to the TGA measurement of SCB in Figure 6, a first mass loss was observed at around 100 °C, with a 7.5% loss corresponding to the water evaporation, which was consistent with the DM determined. The second mass loss was measured between 190 °C and 350 °C, with a 53% mass loss. This could be attributed to cellulose and hemicellulose thermal degradation. According to the literature, hemicellulose degrades first [46] at temperatures of 200 °C to 300 °C and then cellulose degrades at slightly higher temperatures of 300 °C to 350 °C [47]. However, it has been reported that cellulose started to depolymerize at temperatures below 300 °C and the rate of depolymerization was dependent on the applied temperature [48]. Above 350 °C, a third mass loss of 20% was observed. This could be attributed to the thermal degradation of lignin between 350 °C and 500 °C [49].

Binderless SCB materials became darker as the molding temperature increased (Figure 3) and their chemical composition at different molding temperatures are shown in Figure 7. There were no real changes in the chemical composition with the molding temperature, but the material obtained at 240 °C had a higher water-soluble extractable content. It has been shown that increasing the temperature from 180 °C to 240 °C should increase furfural release [50]. In our case, the water-soluble extractable content increase in the materials could have been linked to hemicellulose degradation into residual sugars, which could in turn be converted into sugar-degradation products, such as furfural. Furfural bound in the extractables could have reacted inside the materials, thereby contributing to the self-adhesion [51], and the darkening of the materials could then have been linked to crosslinking reactions occurring during high-temperature thermocompression between furfural and lignin or in the lignin–carbohydrate complex (LCC).

Our results on variations in the material density, mechanical properties and water resistance of binderless SCB materials with different molding temperatures are presented in Figure 8. Molding temperature increases had a densification effect on the material, with a linear increase in the material density from 1308 to 1487 kg/m^3^. When temperature is associated with high pressure during compression, the specific surface area of cellulosic materials falls drastically [52], indicating partial interdiffusion of the external parts of the fibers, which was responsible for the higher densities measured due to a loss of interparticle porosity. The mechanical properties improved from 1.1 to 6.1 GPa for the flexural modulus and from 7.9 to 39.8 MPa for the flexural strength at breaking point. The mechanical properties clearly tended to increase with increasing temperature and reached maximum values at 200 °C, with 6.0 GPa for the flexural modulus and 39.8 MPa for the flexural strength at breaking point. Unlike the pattern observed for the molding pressure, with material densities increasing beyond 180 °C, the mechanical properties no longer increased with temperatures above 200 °C. Generally, a rise of molding temperature leads to higher mechanical properties [53], but an optimum temperature exists due to thermal degradation of the lignocellulosic material [54]. In our case, thermal degradation of SCB constituents in the material composition was not clearly visible. The plateau observed with regard to the mechanical properties could have been linked to hemicellulose plasticization and lignin softening [55] at high temperature. For water resistance, the three specimens prepared at 60 °C and 80 °C and one specimen at 100 °C and 120 °C deteriorated before the end of the 24 h test in water. A minimum temperature of 140 °C was required to ensure the cohesion of the material after immersion in water for 24 h. For the other materials, WA ranged from 32% to 240% and TS from 33% to 208%. WA and TS decreased with temperature and reached minimum values at 240 °C, with 32% WA and 33% TS. As in the case of the molding pressure, the water-resistance enhancement was correlated with the material density. In contrast to the mechanical properties, the water-resistance properties continued to improve beyond 200 °C, which means that lignin and hemicellulose plasticization at high temperatures could be beneficial for water resistance. Plasticization of lignocellulosic compounds could have improved their creep and mobility in the material during thermocompression, thereby promoting the self-adhesion mechanisms and bonding between particles in the binderless material, in turn limiting water diffusion through the material [20].

#### 3.2.3. Effect of Molding Time

The SCB materials became darker with molding time (Figure 3), indicating chemical changes in the material composition, as shown in Figure 9. The hemicellulose content decreased, while the amount of the extractables increased with the molding time. Increasing the molding time appeared to result in hemicellulose hydrolysis, leading to soluble compound releases. These findings were in accordance with those of a study on kenaf core binderless particleboard [16]. As for the high temperatures, sugar-degradation products could react inside the material, thus contributing to self-adhesion. One of the most common self-adhesion mechanisms described in the literature starts with the formation of sugar-degradation compounds, such as HMF and furfural by hydrolysis of hemicellulose into xyloses, followed by xylose dehydration and cellulose degradation during heat treatment [56]. Crosslinking reactions could then occur in the lignin network with the involvement of sugar-degradation products.

Variations in the material density, mechanical properties and water resistance of the materials with molding time are presented in Figure 10. Unlike the molding pressure and temperature, the molding time had a limited impact on the material density, ranging only from 1460 to 1502 kg/m^3^, and on the flexural properties, with the modulus ranging from 4.9 to 6.1 GPa and the flexural strength from 27.5 to 40.8 MPa. Increasing the molding time from 5 to 10 min improved the mechanical properties (Figure 5b). These results were in accordance with those of a previous study on SCB materials [53]. A short molding time did not seem sufficient to transfer heat throughout the material and it limited between-particle bonding [42,57]. However, a decline in the mechanical properties was observed above 15 min. Hemicellulose hydrolysis could have caused deterioration of the structure and self-strength of the materials [21], and cellulose could have start to depolymerize, resulting in lower mechanical properties, as observed with the long molding time. It seemed that an intermediate time was sufficient to obtain the highest mechanical properties, as reported in a previous study [29]. For water resistance, all materials were able to withstand 24 h water immersion, with WA ranging from 21% to 125% and TS from 9% to 123%. WA and TS decreased with molding time and the material with the best water resistance was obtained at 30 min (21% WA and 9% TS). Increasing the molding time from 5 to 10 min led to a three-fold reduction in WA and TS, which could be explained by better bonding between particles and possible internal chemical reorganization by crosslinking reactions during thermocompression, thus limiting water penetration into the material. Beyond 10 min, the decrease in WA and TS continued, although at a slower rate, and could be explained by the internal chemical reorganization following hemicellulose hydrolysis. Hemicellulose, because of its hydrophilic character, is one of the main causes of dimensional instability in binderless lignocellulosic materials. A reduction in the hemicellulose content could promote a decrease in hydroxyl groups and water absorption sites, thereby leading to less water absorption and cell wall thickness swelling and increased dimensional stability of the material [38]. A correlation between the hemicellulose content and TS and WA of the materials made with *Miscanthus sinensis* was also noted by Velásquez et al. [19]. The presence of a higher extractable content, mainly composed of residual free sugars derived from hemicellulose hydrolysis in the water-soluble extractables, and hydrophobic molecules in the ethanol-soluble extractables, could act as a natural adhesive for bonding and contribute to the improvement of water resistance of the materials. The presence of more sugar extractables and their positive effects on TS was mentioned in a previous study on wood species [58].

#### 3.2.4. Effect of the Fiber/Fine-Particle Ratio

The chemical composition of binderless SCB materials obtained with different fiber/fine-particle ratios is shown in Figure 11. As described previously in Table 3, the fiber and fine-particle fractions had different chemical compositions, with higher ash and extractable contents and lower cellulose and hemicellulose contents for the fine-particle fraction. The variations in the material compositions therefore depended on the starting fraction compositions and proportions. Unlike what we observed with regard to the molding time, the modification in the number of extractable molecules in the materials was not due to the thermocompression process, but rather to the fact that the fine particles had a higher proportion of these extractables at the beginning of the process. In addition, materials made with a higher proportion of fine particles had smoother and darker surfaces than those made with a high proportion of fibers (Figure 3), due to the differences in composition and better curing.

The results on variations in the material density, mechanical properties and water resistance with the fiber content are shown in Figure 12. The density and mechanical properties increased slightly with the fine-particle content, and with a linear correlation, from 1437 to 1507 kg/m^3^ for the density, 4.4 to 6.9 GPa for the flexural modulus and 32.3 to 43.6 MPa for the flexural strength. These results are in accordance with those of previous studies on binderless particleboard prepared from SCB [30], unstalked cotton [59] and raw palm-based lignocellulosic biopolymers [60]. Fine particles had a higher specific surface area (Table 3), which improved the accessibility to inner cell wall components [61]. The use of fine particles led to a larger area of contact and reactivity between particles, resulting in an internal chemical reorganization during thermocompression and better self-adhesion of the material with more interparticle bonding [62,63]. Regarding the effects of the hemicellulose content on the materials (15–22%), a lower hemicellulose content seemed to lead also to a material with better mechanical properties.

With regard to water resistance, all materials were able to withstand the 24 h water immersion. WA ranged from 51% to 204% and TS from 47% to 211% and both increased with the fiber content with a polynomial correlation. The fiber fraction was richer in cellulose, which swelled more easily with water, causing the cellulose chains to spread further apart, thereby leaving more space for water absorption. A linear correlation was established between WA and the cellulose content by varying the fiber/fine-particle ratio (Figure 13a). In addition, the fine-particle fraction filled the void volume between fibers, creating a much denser system that prevented water from seeping through the material by increasing the compression ratio [57], which led to a decrease in TS [64]. As noted regarding the molding time, the presence of a lower hemicelluloses content and a higher extractable content (13% in total), mainly composed of residual sugars, in the fine-particle fraction may also have contributed to the improvement in water resistance. Self-polymerization or crosslinking reactions involving extractables during thermocompression could provide further protection against water and prevent water from diffusing and distorting or breaking the intermolecular bonds of the material. A linear correlation was also noted between WA and the hemicellulose content by varying the fiber/fine-particle ratio (Figure 13b).

#### 3.2.5. Effect of Moisture Content

The binderless SCB materials became darker as MC increased (Figure 3), indicating a modification in the chemical composition of the materials during thermocompression, as indicated in Figure 14. The increased MC resulted in a decrease in the cellulose and hemicellulose contents, involving a slight increase in ash content and an artificial increase in the lignin content, possibly due to the dosage of the released UV-absorbing compounds. These results implied that there was an acceleration in the sugar-degradation reactions in the presence of higher quantities of water: under the pressure and temperature conditions of our study, water could act as a weak acid and prompt the hydrolysis of lignocellulosic compounds.

Variations in the material density, mechanical properties and water resistance with MC are shown in Figure 15. The material density ranged from 1299 to 1460 kg/m^3^, the flexural modulus from 0.5 to 5.2 GPa and the flexural strength from 3.8 to 36.5 MPa. The material density and mechanical properties had the same variation pattern with MC: these properties increased from 0 to 7.2% MC and then decreased when MC was above 10%. However, the lowest values were obtained with materials made with dry SCB and two of these tested samples were deteriorated before the end of 24 h of water immersion. In line with our results, it has been reported that dried material molding is generally unsuccessful [30]. The poor mechanical properties observed with <7.2% MC could have been linked to the effect of water on the lignin glass transition temperature: a correlation was established by Kelley et al., showing that the lignin glass transition temperature of the wood decreased with <10% MC [65]. Materials should be manufactured with a MC that allows them to exceed the lignin glass transition temperature so as to promote its mobilization and plasticization, thereby improving self-bonding of the material and its final properties [66]. The improvement in mechanical properties observed when MC increased from 0 to 7.2% could be also explained by the higher material density obtained, the improvement in heat transfer [47], the contribution of water to bonding via hydrogen bonding [21] and capillary sorption between particles among the fibers [66]. Increasing MC to >7.2% led to a steady decline in the mechanical properties, in accordance with previous results [23,67]. This could have been due to many factors such as water adsorption into the cell walls, causing swelling and alteration of the lignocellulosic structure, the incompressibility of wet biomass particles, since water is incompressible, thus causing a reduction in the contact between fibers [67] or a plasticizing effect of water, thereby facilitating deformation of the material.

With regard to water resistance, WA ranged from 72% to 137% and TS from 46% to 133% after 24 h of immersion. WA and TS decreased with MC before reaching a plateau at 12.4% MC for WA and 15.4% MC for TS. A high MC could cause internal chemical reorganization during thermocompression, linked to hemicellulose and cellulose degradation by water, thus making the final material more water-resistant. In addition, it has been reported that WA and TS decreased drastically with MC due to a lower lignin softening point and better heat transfer, but that effect was attenuated when the lignin softening temperature was exceeded [15]. This could explain the plateau observed from 12.4% MC for WA, as the lignin softening point could have been reached. Another explanation could be related to the quantity of the bound water. Sui et al. described that when water is poured into lignocellulosic fiber, the water is first located inside the cell wall matrix as bound water, thereby increasing the cell wall mass and volume and swelling of the samples [68]. In this study, the sample properties declined with the MC when the system reached equilibrium, with the liquid water fraction beginning to accumulate in the voids as free water, while the cell walls stopped swelling and their properties no longer changed with the MC. The plateau observed for WA and TS from 12.4% MC could therefore have corresponded to the equilibrium at which the maximum amount of bound water had been reached.

### 3.3. Choice of Best Conditions

The material with the highest mechanical properties in this study was prepared with only fine particles, with a flexural modulus of 6.9 GPa and a flexural strength of 43.6 MPa. In comparison, other studies on manufacturing binderless materials from bagasse reported much lower mechanical properties (flexural modulus < 2 GPa and flexural strength < 20 MPa) [26,29,53,69]. Meanwhile, the material with the highest water resistance was prepared with a 30 min molding time, giving a WA of 21% and a TS of 9%. Under these conditions, the material had the lowest hemicellulose content (13%) and the highest amount of total extractable elements (16%) after thermocompression. Nadhari et al. obtained similar WA (13%) and TS (10%) levels for binderless bagasse particleboard, but their mechanical properties were lower (1.7 GPa for the flexural modulus and 15.2 MPa for the flexural strength) [29]. Other studies on binderless materials from bagasse reported higher TS values of 43 to 93% [26,53,69]. 

The best trade-off to obtain a material with acceptable mechanical properties and water resistance was thus determined: the material prepared with only fine particles had the highest mechanical properties and acceptable water resistance with a WA of 51% and TS of 47%. This choice of operating conditions was more relevant from a practical viewpoint given that a 30 min molding time was relatively long to implement. Another possibility was to use a 10 min molding time in order to reduce the molding time by three-fold while keeping acceptable mechanical properties (6 GPa for the flexural modulus and 41 MPa for the flexural strength) and achieving better water resistance (45% for WA and 41% for TS) than the material obtained with only fine particles, which only represented 35% of the initial ground SCB. Under these conditions, the mechanical properties of the materials obtained were superior to those of materials obtained with bagasse containing binders (flexural modulus < 4 GPa and flexural strength < 30 MPa), and WA was also lower or equivalent (29–93%), but TS was higher compared to that of materials with binders (4–32%) [38,42,46,70].

Based on the properties measured, the materials obtained could be used as panels for furniture or building applications. According to standard NF EN 312 which described the requirements for particleboard made from wood fibers [71], the materials obtained exhibited properties that meet the criteria for P2 type panels, corresponding to panels used for interior fittings and furniture in a dry environment. For outdoor use, TS value had to be less than 25% for P3 type panels, which eliminated most of the materials, but the material obtained with a 30 min molding time met this criterion.

## 4. Conclusions

100% lignocellulosic binderless materials using SCB were manufactured by high-temperature uniaxial compression to investigate the effects of the moisture content, fiber/fine-particle ratio and molding conditions on their properties and chemical compositions. The properties of the materials were affected by all parameters. Overall, the materials had better mechanical properties when SCB was thermocompressed at high pressure (≥68 MPa), high temperature (≥200 °C) and an intermediate time (10 min), and when fine particles were used, due to the densification of the materials and better bonding between particles. The WA and TS of the materials were improved with higher temperature (240 °C), time (30 min) and moisture content. This was a result possibly due to densification, lignin plasticization and internal chemical reorganization during thermocompression with crosslinking reactions in the LCC network, or to involving sugar-degradation products or extractables derived from hemicellulose hydrolysis. The chemical compositions of the final materials, i.e., specifically cellulose, hemicellulose and extractable contents, were affected by the molding time and moisture content used, but also by the fiber/fine-particle ratio chosen before thermocompression. The cellulose and hemicellulose contents in the materials decreased with the moisture content and hemicellulose was degraded into extractable molecules with a long molding time. 

## Figures and Tables

**Figure 1 materials-17-01713-f001:**
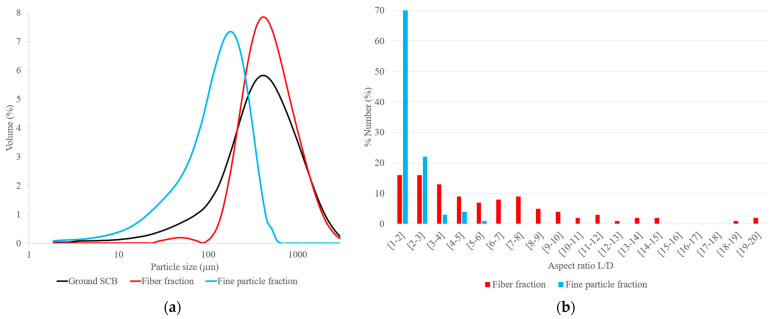
(**a**) Particle-size distribution of ground SCB and its two granulometric fractions (fiber and fine particle); (**b**) aspect ratio (L/D) of the fiber and fine-particle fractions.

**Figure 2 materials-17-01713-f002:**
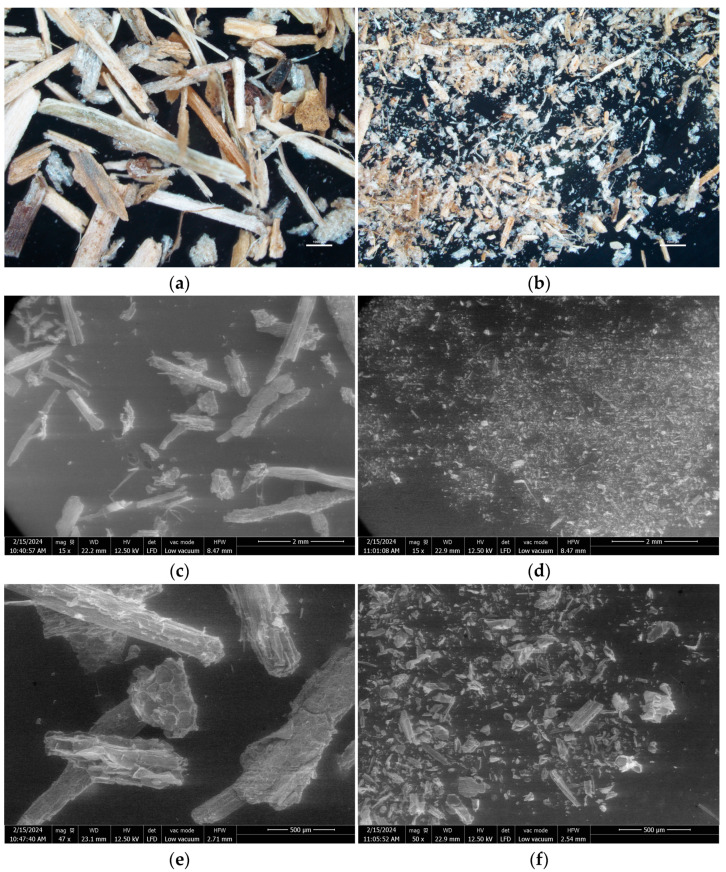
Binocular view of (**a**) the fiber fraction, and (**b**) the fine-particle fraction; SEM images of the fiber fraction at (**c**) ×15 and (**e**) ×50 magnification and of the fine-particle fraction at (**d**) ×15 and (**f**) ×50 magnification, obtained from ground SCB.

**Figure 3 materials-17-01713-f003:**
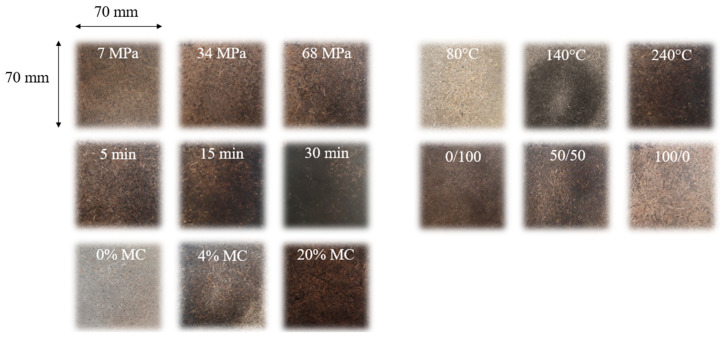
Pictures of the SCB materials obtained at different pressures, temperatures, times, fiber/fine-particle ratios and moisture content.

**Figure 4 materials-17-01713-f004:**
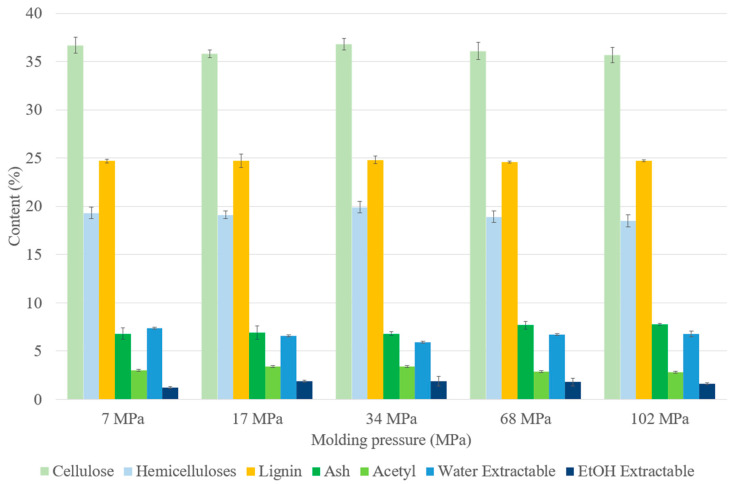
Chemical composition on a dry-matter basis of binderless SCB materials at different molding pressures (Table 1: 180 °C, 5 min, 7.2% MC, initial ground SCB). Errors bars represent the standard deviation.

**Figure 5 materials-17-01713-f005:**
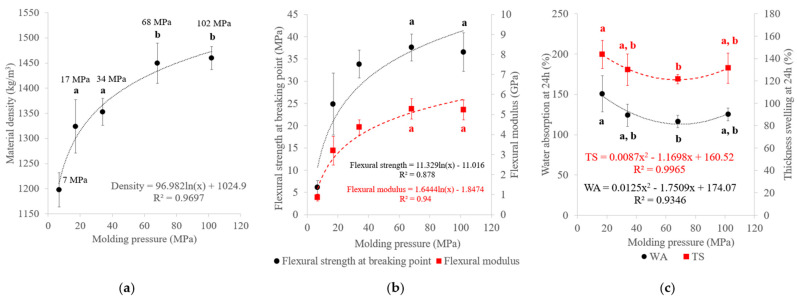
Variations in (**a**) material density, (**b**) flexural properties and (**c**) water-resistance properties of binderless SCB materials with molding pressure (Table 1: 180 °C, 5 min, 7.2% MC, initial ground SCB). a,b letters on the graphs refer to Student’s *t* test results and error bars represent the standard deviation.

**Figure 6 materials-17-01713-f006:**
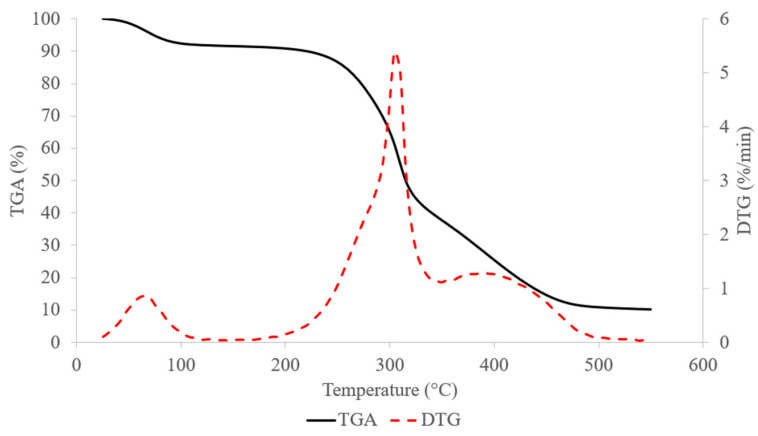
TGA and DTG curves from ground SCB degradation with 5 °C/min airflow.

**Figure 7 materials-17-01713-f007:**
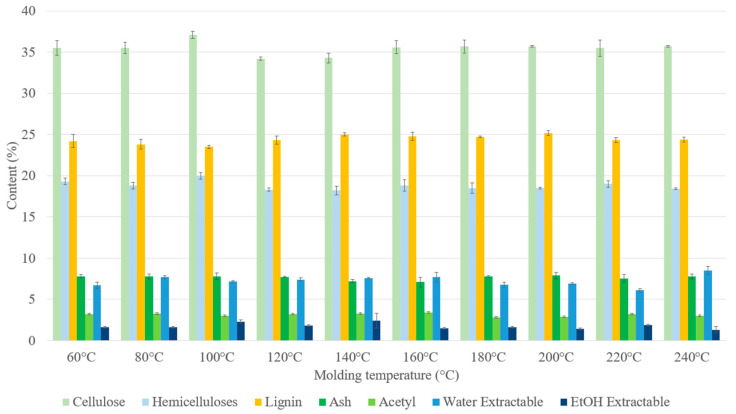
Chemical composition on a dry-matter basis of binderless SCB materials at different molding temperatures (Table 1: 102 MPa, 5 min, 7.2% MC, initial ground SCB). Errors bars represent the standard deviation.

**Figure 8 materials-17-01713-f008:**
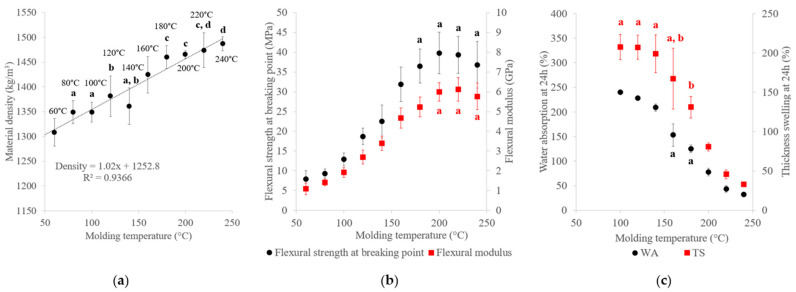
Variations in the (**a**) material density, (**b**) flexural properties and (**c**) water-resistance properties of binderless SCB materials with molding temperature (Table 1: 102 MPa, 5 min, 7.2% MC and a fiber/fine-particle ratio of 65/35). a–d letters on the graphs refer to Student’s *t* test results and error bars represent the standard deviation.

**Figure 9 materials-17-01713-f009:**
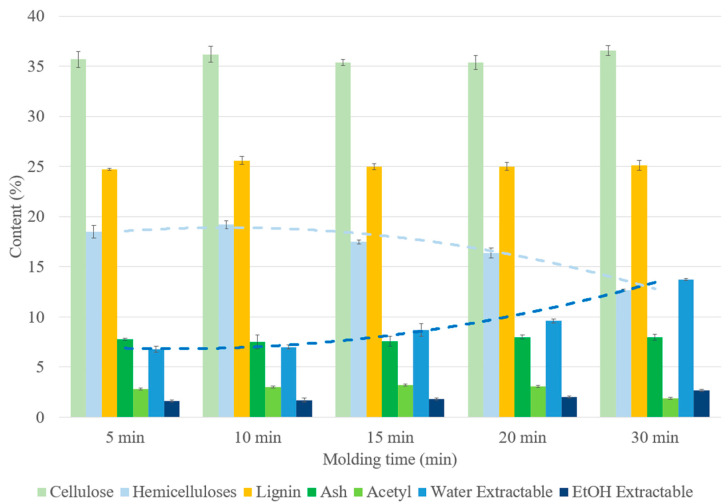
Chemical composition on a dry-matter basis of binderless SCB materials at different molding times (Table 1: 102 MPa, 180 °C, 7.2% MC, initial ground SCB). Errors bars represent the standard deviation and the light and dark blue dashed lines correspond to hemicelluloses and water extractable contents, respectively.

**Figure 10 materials-17-01713-f010:**
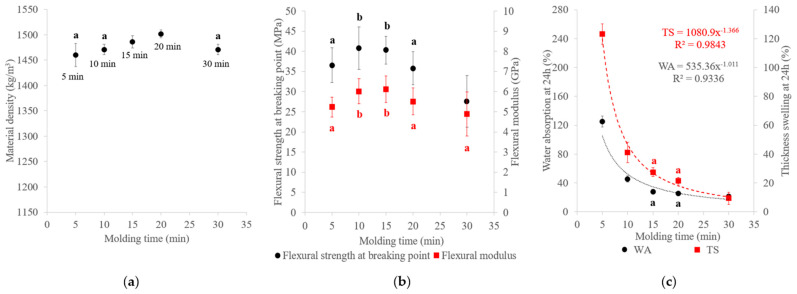
Variations in the (**a**) material density, (**b**) flexural properties and (**c**) water-resistance properties of binderless SCB materials with molding time (Table 1: 102 MPa, 180 °C, 7.2% MC, initial ground SCB). a,b letters on the graphs refer to Student’s *t* test results and error bars represent the standard deviation.

**Figure 11 materials-17-01713-f011:**
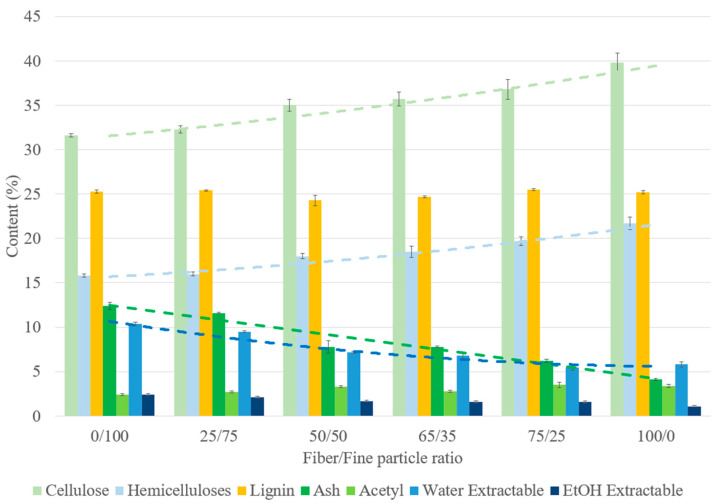
Chemical composition on a dry-matter basis of SCB binderless materials at different fiber/fine-particle ratios (Table 1: 102 MPa, 180 °C, 5 min, 7.2% MC). Errors bars represent the standard deviation. The light and dark green dashed lines correspond to cellulose and ash contents, respectively and the light and dark blue dashed lines correspond to hemicellulose and water extractable content, respectively.

**Figure 12 materials-17-01713-f012:**
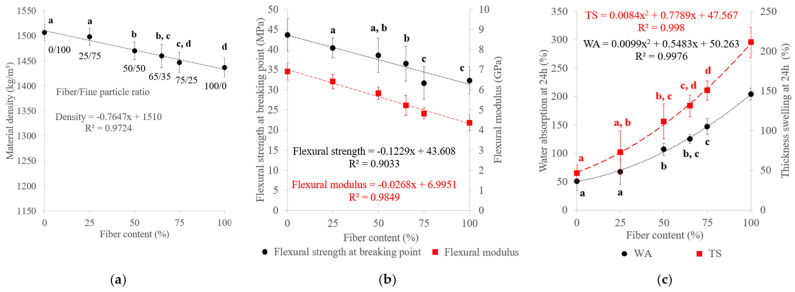
Variations in the (**a**) material density, (**b**) flexural properties and (**c**) water-resistance properties of binderless SCB materials with the fiber content (Table 1: 102 MPa, 180 °C, 5 min, 7.2% MC). a–d letters on the graphs refer to Student’s *t* test results and error bars represent the standard deviation.

**Figure 13 materials-17-01713-f013:**
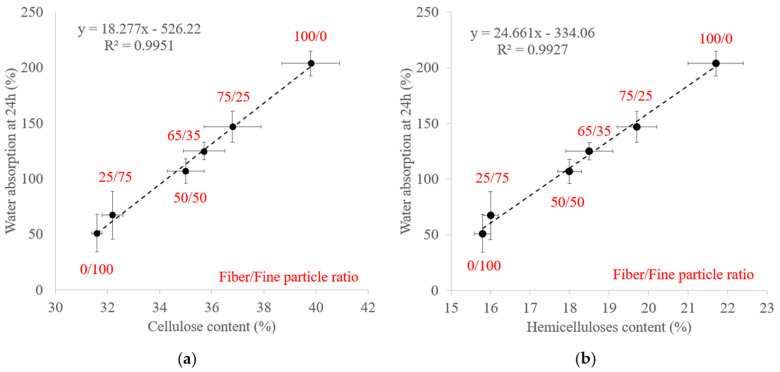
Variations in water absorption after 24 h immersion with the (**a**) cellulose content and (**b**) hemicellulose content by varying the fiber/fine-particle ratio. Errors bars represent the standard deviation.

**Figure 14 materials-17-01713-f014:**
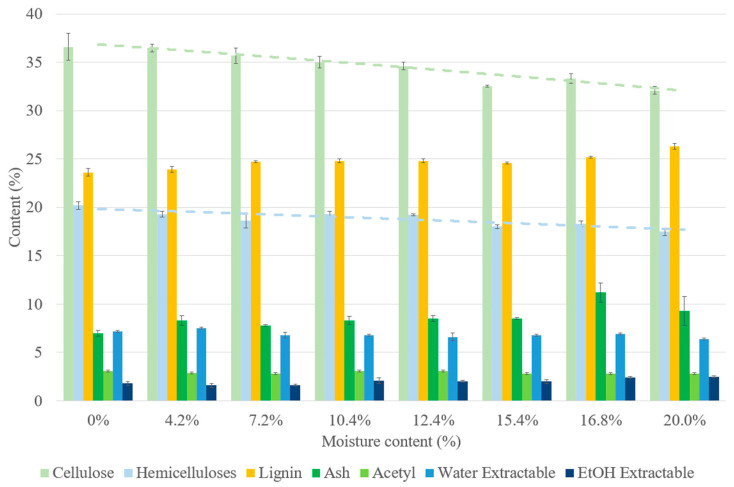
Chemical composition on a dry-matter basis of binderless SCB materials at different moisture contents (Table 1: 102 MPa, 180 °C, 5 min, initial ground SCB). Errors bars represent the standard deviation and the light blue and light green dashed lines correspond to cellulose and hemicelluloses contents, respectively.

**Figure 15 materials-17-01713-f015:**
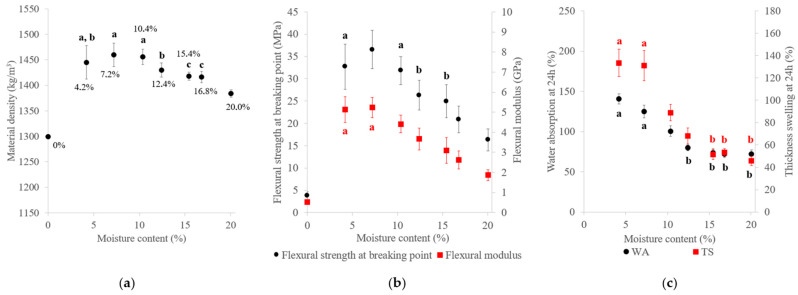
Variations in the (**a**) material density, (**b**) flexural properties and (**c**) water-resistance properties of binderless SCB materials with MC (Table 1: 102 MPa, 180 °C, 5 min and a fiber/fine-particle ratio of 65:35). a–c letters on the graphs refer to Student’s *t* test results and error bars represent the standard deviation.

**Table 1 materials-17-01713-t001:** Thermocompression conditions used for the production of SCB binderless materials.

Studied Parameter	Pressure (MPa)	Temperature (°C)	Time (min)	Fiber/Fine- Particle Ratio (%)	Moisture Content (%)
Pressure	7–102	180	5	65/35	7.2
Temperature	102	60–240	5	65/35	7.2
Time	102	180	5–30	65/35	7.2
Fiber/Fine-particle ratio	102	180	5	0/100–100/0	7.2
Moisture content	102	180	5	65/35	0–20.0

**Table 2 materials-17-01713-t002:** Results on densities, SSA, D10, D50 and D90 of ground SCB and its fiber and fine-particle fractions. Values are reported with their standard deviation.

	Ground SCB	Fibers	Fine Particles
Bulk density (kg/m^3^)	132 ± 3	118 ± 1	138 ± 4
Tapped density (kg/m^3^)	198 ± 7	176 ± 2	208 ± 4
SSA (m^2^/g)	39 ± 3	15 ± 1	98 ± 8
D10 (µm)	106 ± 28	226 ± 2	34 ± 3
D50 (µm)	423 ± 60	490 ± 4	145 ± 11
D90 (µm)	1237 ± 114	1210 ± 28	161 ± 12

**Table 3 materials-17-01713-t003:** Chemical composition on a dry-matter basis of ground SCB and its two fractions (fibers and fine particles). Values are reported with their standard deviation.

Composition	Ground SCB	Fibers	Fine Particles
Dry matter (%)	91.5 ± 0.1	91.3 ± 0.1	92.7 ± 0.1
Ash (%)	7.0 ± 0.4	4.1 ± 0.1	12.3 ± 0.8
AIL (%)	20.4 ± 0.4	19.8 ± 0.7	21.6 ± 1.6
ASL (%)	4.6 ± 0.1	5.1 ± 0.1	4.1 ± 0.2
Total lignin (%)	24.9 ± 0.5	24.9 ± 0.6	25.2 ± 1.6
Cellulose (%)	36.0 ± 0.8	39.5 ± 0.4	31.6 ± 0.3
Xylose (%)	18.3 ± 0.6	21.0 ± 0.3	14.8 ± 0.1
Arabinose (%)	1.5 ± 0.2	1.5 ± 0.1	1.4 ± 0.2
Hemicellulose (%)	19.9 ± 0.5	22.5 ± 0.3	16.2 ± 0.5
Acetyl (%)	2.9 ± 0.1	3.2 ± 0.1	2.4 ± 0.1
Extractable H_2_O (%)	7.1 ± 0.1	4.4 ± 0.6	11.7 ± 0.3
Extractable EtOH (%)	1.1 ± 0.1	1.1 ± 0.1	1.1 ± 0.1
Total (%)	99	99	100

## Data Availability

The raw data supporting the conclusions of this article will be made available by the authors on request.

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
