# Peer review of "Influence of Thermocompression Conditions on the Properties and Chemical Composition of Bio-Based Materials Derived from Lignocellulosic Biomass"

_materials, 2024, doi:10.3390/ma17081713_

Round 1

Reviewer 1 Report

Comments and Suggestions for Authors

Dear authors,

I congratulate you on the work done, however, here are the suggestions/guidelines for changes to contribute to the work to be published.

Line 58 - It is not necessary to mention the author throughout the text. I suggest rewriting the sentence with only the citation numbering;

Line 66 - In order to verify the truthfulness of the statement that there are few works in this line of research, cite the few works.

Line 73 - 79: Do not include methodology at the end of the introduction. Limit yourself to the objective of the study.

Throughout the text, separate °C from the number.

Line 150 Table 1 - I suggest that in the text the evaluation points regarding pressure, temperature, etc., be informed. These pieces of information were only reported in the results and discussion. Also, inform the reason for fixing the exact values of Pressure at 102 MPa, Temperature at 180 °C, Time at 5 min; Fiber/Fine at 65/35; MC at 7.2%.

Line 167 Figure 1 is a result and not methodology. This statement is based on the frequency with which it was mentioned in the Results and Discussion.

Line 206 - Do not use the word "All" at the beginning of the sentence. It gives the impression that it is an acronym. The regression used in Figure 12 and Figure 13 was not mentioned. Regarding the statistical analyses, in order to make any affirmative comparison between the “3.1. Characterization of raw sugarcane bagasse”;

 (Line 212); Table 3. (Line 264)3.2.1. Effect of molding pressure (line 267); 3.2.2. Effect of molding temperature (line 301); 3.2.3. Effect of molding time (line 362); 3.2.4. Effect of the fiber/fine particle ratio (line 412); 3.2.5. Effect of moisture content (line 464); it is necessary to perform a statistical comparison test.

To validate the statements regarding Figures 5, 8, and 10, it is necessary to perform a regression analysis.

Line 242 - It might be interesting to better evaluate this figure, as it does not seem to have spherical particles.

Line 274 Figure 4 - This information "180°C, 5 min, 7.2% MC, initial ground SCB" should be referenced throughout the text referring to Table 2. This observation applies to all figures with this information (Line 325 Figures 7, 8, 9, 10, 11, 12, 14, 15).

Line 315 Figure 6 - The term "DTG" was not mentioned throughout the manuscript. Line 442 Figure 12 - The regression equation was not mentioned.

Lines 546 to 563: In my view, the conclusion seems like an extension of topic 3.3

Line 519. I request that they be more conclusive, responding to the purpose described in the objective of your work.

Author Response

Dear reviewer,

Thank you very much for taking the time to review this manuscript. Please find the detailed responses below and the corresponding revisions highlighted in the re-submitted files.

Comment 1: Line 58 - It is not necessary to mention the author throughout the text. I suggest rewriting the sentence with only the citation numbering.

Response 1: Thank you for this comment. We have modified the sentence, accordingly, in line 63 in the revised version.

Comment 2: Line 66 - In order to verify the truthfulness of the statement that there are few works in this line of research, cite the few works.

Response 2: Thank you for pointing this out. We have cited two references in this line of research in lines 73.

Comment 3: Line 73 - 79: Do not include methodology at the end of the introduction. Limit yourself to the objective of the study.

Response 3: Thank you for you comment. We have modified this paragraph and deleted these lines.

Comment 4: Throughout the text, separate °C from the number.

Response 4: Thank you for pointing this out. We agree and, accordingly, we have made the modifications in the revised version.

Comment 5: Line 150 Table 1 - I suggest that in the text the evaluation points regarding pressure, temperature, etc., be informed. These pieces of information were only reported in the results and discussion. Also, inform the reason for fixing the exact values of Pressure at 102 MPa, Temperature at 180 °C, Time at 5 min; Fiber/Fine at 65/35; MC at 7.2%.

Response 5: Thank you for pointing this out. We agree with this comment and, accordingly, we have revised the manuscript with the mention directly in the text of the evaluation points in lines 165 to 168. We have also developed the reasons for fixing the exact values in the same section in lines 169 to 176.

Comment 6: Line 167 Figure 1 is a result and not methodology. This statement is based on the frequency with which it was mentioned in the Results and Discussion.

Response 6: Thank you for pointing this out. We have, accordingly, revised the manuscript and this figure is now presented in the results part in line 303 (Figure 3).

Comment 7: Line 206 - Do not use the word "All" at the beginning of the sentence. It gives the impression that it is an acronym. The regression used in Figure 12 and Figure 13 was not mentioned. Regarding the statistical analyses, in order to make any affirmative comparison between the “3.1. Characterization of raw sugarcane bagasse”; (Line 212); Table 3. (Line 264)3.2.1. Effect of molding pressure (line 267); 3.2.2. Effect of molding temperature (line 301); 3.2.3. Effect of molding time (line 362); 3.2.4. Effect of the fiber/fine particle ratio (line 412); 3.2.5. Effect of moisture content (line 464); it is necessary to perform a statistical comparison test.

Response 7: Thank you for this comment. We have removed the word “All” in the sentence mentioned in line 234 in the revised version. According to your comment, we have mentioned also the determination of the regression when it was possible in line 238. For the statistical analyses, we used ANOVA analysis to verify if there were any significantly different values, and the Student’s test to differentiate the values obtained for each thermocompression parameter evaluated.

Comment 8: To validate the statements regarding Figures 5, 8, and 10, it is necessary to perform a regression analysis.

Response 8: Thank you for this comment. Accordingly, we have performed regression analysis and we have added the corresponding regression in the Figures 5, 8 and 10 when R² was higher than 0.85.

Comment 9: Line 242 - It might be interesting to better evaluate this figure, as it does not seem to have spherical particles.

Response 9: Thank you for this comment. Fiber fraction contains only a small proportion of spherical particles, according to the measurement of the aspect ratio with 15% of the particles with a L/D between 1 and 2. The long fibers also tended to mask the spherical particles, but it is possible to see an example of rather spherical particles in the fiber fraction in the SEM image in Figure 2e.

Comment 10: Line 274 Figure 4 - This information "180°C, 5 min, 7.2% MC, initial ground SCB" should be referenced throughout the text referring to Table 2. This observation applies to all figures with this information (Line 325 Figures 7, 8, 9, 10, 11, 12, 14, 15).

Response 10: Thank you for this comment. We agree with that and we have referenced the Table in  all the figure captions mentioned (Table 1 in the revised version).

Comment 11: Line 315 Figure 6 - The term "DTG" was not mentioned throughout the manuscript.

Response 11: Thank you for pointing this out. We revised the manuscript to mention the DTG term in the line 157.

Comment 12: Line 442 Figure 12 - The regression equation was not mentioned.

Response 12: Thank you for pointing this out. We revised the manuscript and we added the equation directly in the figure (Line 480)

Comment 13 : Lines 546 to 563: In my view, the conclusion seems like an extension of topic 3.3. Line 519. I request that they be more conclusive, responding to the purpose described in the objective of your work.

Response 13 : Thank you for your comment. In the topic 3.3, we have tried to present the best conditions to obtain the highest properties of the materials and to compare them with similar materials prepared using bagasse. Following the others revisions, we have added a short paragraph on the potential industrial applications and comparison with particleboard made from wood fibers. In the conclusion, we have instead tried to summarise the different effects of thermocompression parameters on material properties, while indicating the conditions to be favoured for each parameter in order to obtain high properties. We have also mentioned the effects of these parameters on the chemical composition of the materials, as well as the phenomena involved, such as densification, plasticization and internal chemical reorganisation, in order to answer the objective of determining the effects of thermocompression conditions on the properties and chemical composition of the materials.

Reviewer 2 Report

Comments and Suggestions for Authors

An article was submitted for evaluation regarding the use of lignocellulosic particles derived from sugar cane bagasse for the production of porous materials without the use of an adhesive joint.

The article presents the influence of various technological parameters and particle sizes on the properties of the obtained materials - bending strength, flexural modulus, swelling under the influence of water.

The article is well prepared and the authors of the publication carefully prepared the research plan.

What was missing in the article, but could be presented in the introduction or discussion of the results, is what industrial/utility applications such materials can be used for. Can the obtained materials successfully replace materials made of wood fibers, such as soft board or low denisty board? Can the authors indicate whether the characterized materials have better, worse or similar parameters than materials based on wood particles?

It is a pity that the authors of the research did not support their results with tomographic tests that allowed to determine the degree of porosity of the materials produced and tests to assess the crystallinity of cellulose. Therefore, I encourage the authors of the publication to continue research in this area.

Technical notes:

Please standardize the Y axis in Figures 15.

Were the SEM samples saturated with gold? The photos show that probably not. SEM images would be clearer.

Author Response

Dear reviewer,

Thank you very much for taking the time to review this manuscript. Please find the detailed responses below and the corresponding revisions highlighted in the re-submitted files.

Comment 1: What was missing in the article, but could be presented in the introduction or discussion of the results, is what industrial/utility applications such materials can be used for. Can the obtained materials successfully replace materials made of wood fibers, such as soft board or low denisty board? Can the authors indicate whether the characterized materials have better, worse or similar parameters than materials based on wood particles?

Response 1: Thank you for your comment. We have therefore revised the manuscript to highlight this point by mentioning the potential applications of these materials in the topic 3.3. Choice of best conditions in the lines 588 to 594. We have compared these materials to standard particleboard to assess whether these materials could be used to replace industrial particleboard.

Comment 2: It is a pity that the authors of the research did not support their results with tomographic tests that allowed to determine the degree of porosity of the materials produced and tests to assess the crystallinity of cellulose. Therefore, I encourage the authors of the publication to continue research in this area.

Response 2: Thank you for your advice and encouragement. We are continuing our research in this area and plan to carry out new characterizations to further evaluate these materials.

Comment 3: Please standardize the Y axis in Figures 15.

Response 3: Thank you for your comment. We have modified the figures and the Y axis is now standardized (line 542).

Comment 4: Were the SEM samples saturated with gold? The photos show that probably not. SEM images would be clearer.

Response 4: Thank you for pointing this out. The samples were not saturated with gold, we will take it into account for our next SEM analysis and we have mentioned it in line 154 in the revised version.

Reviewer 3 Report

Comments and Suggestions for Authors

The manuscript is focused on investigation and evaluation of the effect of thermocompression conditions on selected properties and chemical composition of biobased materials derived from lignocellulosic biomass. In this respect, the manuscript is within the scope of the Special Issue “Recent Developments in Bio-Based Particleboards and Fiberboards” in Materials journal. In general, the manuscript is well-developed, structured, and informative, however, it needs some revisions before acceptance for publication.

The title (lines 2-4) and the keywords (lines 26-27) is relevant to the aims and objectives of the manuscript.

Overall, the abstract (lines 11-25) is informative, and contains the main findings of the research work.

Lines 17-18: “Water absorption (WA) and thickness swelling (TS) respectively ranged from 21% to 240% and from 9% to 208%.”: I believe it slightly should be modified to “Water absorption (WA) and thickness swelling (TS) values ranged from 21% to 240% and from 9% to 208%, respectively”.

Lines 30-31: here I would suggest adding also “renewable”.

Line 34: please check this new relevant reference on the use of agroindustrial biomass for production of particleboard panels: https://doi.org/10.1016/j.jmrt.2022.08.166

Lines 37-38: “The use of these chemicals may have a negative impact by giving rise to environmental and health issues.”: the statement is generally true, as the main human health hazard of the commercial synthetic resins used for bonding wood composites is the free formaldehyde emission from the finished panels. Please add some short information in this respect supported by relevant references.

Overall, the Introduction part is well-developed, structured and informative, and provides relevant information on the topic of the research, based on previously research works in the field. The aim of the research is also clearly presented.

Lines 151-153: please explain/justify the values of the selected experimental variables, i.e., pressure, temperature, time, fiber/fine particle ratio and moisture content.

In general, the Materials and Methods section is provides sufficient information on the materials, methods, and equipment used to conduct the experimental work.

Line 236, Figure 2b is not very clear, please replace it with a sharper graph, if possible.

Line 274, Figure 4, and onward in the manuscript: please include in the figure caption that the error bars represent the standard deviation.

In general, the results of the study are properly presented and discussed with relevant research papers in the respective field.

The Conclusion part reflects the main findings of the manuscript. Here I would suggest adding some short information about the practical (industrial) application of the results obtained and the potential for future studies the field.

The references cited are appropriate and correspond to the topic of the manuscript. 

Comments on the Quality of English Language

The English language and style used are fine with only some minor issues detected. 

Author Response

Dear reviewer,

Thank you very much for taking the time to review this manuscript. Please find the detailed responses below and the corresponding revisions highlighted in the re-submitted files.

Comment 1: Lines 17-18: “Water absorption (WA) and thickness swelling (TS) respectively ranged from 21% to 240% and from 9% to 208%.”: I believe it slightly should be modified to “Water absorption (WA) and thickness swelling (TS) values ranged from 21% to 240% and from 9% to 208%, respectively”.

Response 1: Thank you for pointing this out. Therefore, we have modified the sentence according to your comment in line 17.

Comment 2: Lines 30-31: here I would suggest adding also “renewable”.

Response 2: We agree with this comment and we have added “renewable” in the manuscript in the line 31.

Comment 3: Line 34: please check this new relevant reference on the use of agroindustrial biomass for production of particleboard panels: https://doi.org/10.1016/j.jmrt.2022.08.166

Response 3: Thank you for sending us this reference, we have added it to mention more examples of particleboards in line 34.

Comment 4: Lines 37-38: “The use of these chemicals may have a negative impact by giving rise to environmental and health issues.”: the statement is generally true, as the main human health hazard of the commercial synthetic resins used for bonding wood composites is the free formaldehyde emission from the finished panels. Please add some short information in this respect supported by relevant references.

Response 4: Thank you for this comment. We have, accordingly, revised the manuscript to emphasize this point with the mention of  the formaldehyde and volatile organic compounds emissions with two references in lines 38-41.

Comment 5: Lines 151-153: please explain/justify the values of the selected experimental variables, i.e., pressure, temperature, time, fiber/fine particle ratio and moisture content.

Response 5: We agree with this comment and, accordingly, we have developed this point in the lines 169 to 176.

Comment 6: Line 236, Figure 2b is not very clear, please replace it with a sharper graph, if possible.

Response 6: Thank you for pointing this out. We have modified the quality of the figure so it should be clearer (line 267 in the revised version)

Comment 7: Line 274, Figure 4, and onward in the manuscript: please include in the figure caption that the error bars represent the standard deviation.

Response 7: Thank you for this comment. Accordingly, we have added this sentence in all the figure captions.

Comment 8: The Conclusion part reflects the main findings of the manuscript. Here I would suggest adding some short information about the practical (industrial) application of the results obtained and the potential for future studies the field.

Response 8: Thank you for this comment. We have added a short paragraph to describe the potential applications of these materials in the section 3.3. Choice of best conditions in lines 588-594.